# Interventional Oncology and Immuno-Oncology: Current Challenges and Future Trends

**DOI:** 10.3390/ijms24087344

**Published:** 2023-04-16

**Authors:** Alessandro Posa, Andrea Contegiacomo, Francesca Romana Ponziani, Ernesto Punzi, Giulia Mazza, Annarita Scrofani, Maurizio Pompili, Shraga Nahum Goldberg, Luigi Natale, Antonio Gasbarrini, Evis Sala, Roberto Iezzi

**Affiliations:** 1Department of Diagnostic Imaging, Oncologic Radiotherapy and Hematology, Fondazione Policlinico Universitario A. Gemelli IRCCS, L.go A. Gemelli 8, 00168 Rome, Italy; alessandro.posa@policlinicogemelli.it (A.P.); andrea.contegiacomo@policlinicogemelli.it (A.C.); ernesto.punzi@gmail.com (E.P.); giulia.mazza05@icatt.it (G.M.); annarita.scrofani@guest.policlinicogemelli.it (A.S.);; 2Internal Medicine and Gastroenterology-Hepatology Unit, Fondazione Policlinico Universitario A. Gemelli IRCCS, L.go A. Gemelli 8, 00168 Rome, Italy; francescaromana.ponziani@policlinicogemelli.it (F.R.P.); maurizio.pompili@policlinicogemelli.it (M.P.);; 3Facoltà di Medicina e Chirurgia, Università Cattolica del Sacro Cuore, L.go F. Vito 1, 00168 Rome, Italy; 4Division of Image-Guided Therapy, Department of Radiology, Hadassah Hebrew University Medical Center, Jerusalem 12000, Israel; sgoldber@caregroup.harvard.edu

**Keywords:** immunotherapy, interventional oncology, immune checkpoint inhibitors, ablation, personalized medicine

## Abstract

Personalized cancer treatments help to deliver tailored and biologically driven therapies for cancer patients. Interventional oncology techniques are able to treat malignancies in a locoregional fashion, with a variety of mechanisms of action leading to tumor necrosis. Tumor destruction determines a great availability of tumor antigens that can be recognized by the immune system, potentially triggering an immune response. The advent of immunotherapy in cancer care, with the introduction of specific immune checkpoint inhibitors, has led to the investigation of the synergy of these drugs when used in combination with interventional oncology treatments. The aim of this paper is to review the most recent advances in the field of interventional oncology locoregional treatments and their interactions with immunotherapy.

## 1. Introduction

In recent years, we have witnessed the exponential expansion and impact of interventional radiology in oncology. Cross-sectional imaging techniques play a crucial role in the diagnosis, treatment planning, and follow-up of cancer patients, but also provide the ability to perform minimally invasive approaches to procure tissue for histological diagnosis, including the genetic material necessary to develop better tailored and biologically driven treatments. This permits personalized medicine, which potentially maximizes therapeutic effects [1]. Moreover, there is increasing attention in the scientific and medical community on the development of interventional oncology techniques and procedures as locoregional approaches to be employed in cancer treatment in a multidisciplinary cancer management setting.

Currently, percutaneous interventional approaches are being performed for the treatment of a wide range of both primary and secondary malignancies as an alternative or in combination with surgery and other treatment modalities [2]. Indeed, multidisciplinary guidelines for the treatment of HCC and RCC now incorporate their use [3,4,5].

Interventional oncology has the unique capability to treat malignancy in a locoregional fashion, enabling curative (ablative treatments), disease control (intra-arterial chemo- or radio-embolization), and palliative treatment [6]. Locoregional eradication therapy involves the application of different energy sources that, despite a variety of mechanisms of action (such as heat, freezing, or electricity), induce effective necrosis of the tumor core and apoptosis of the adjacent tissue, with a substantial preservation of healthy parenchyma [7].

The destruction of the tumor, along with the release of necrotic material, creates in situ availability of antigens that may be recognized by the immune system as a threat and potentially trigger an immune response throughout the body, bringing to the so-called abscopal effect [8,9].

With the advent of immunotherapy in cancer care and the introduction of immune checkpoint inhibitors, several efforts have been made to investigate the synergy of such treatment in combination with interventional radiology treatments, as this new class of drug influences the immunologic microenvironment of the tumor-acting on several key target molecules, restoring immune system function against malignancy [10].

Although preliminary evidence from immune checkpoint inhibitors monotherapy is promising, the greatest potential of these treatments is likely to be achieved in their combination with other treatments that can trigger an immune response [11]. In this manuscript, we review the most recent advances in locoregional interventional oncology treatments and their interactions with immune checkpoint inhibitors. Figure 1 depicts the most important interventional oncology techniques and their mode of action.

## 2. Percutaneous Treatment and Immuno-Oncology

Based upon multi-disciplinary guidelines, percutaneous interventional techniques are currently considered a possible therapeutic strategy to treat both primary and secondary tumors in multiple anatomical sites [12,13,14,15]. However, therapeutic outcomes of interventional techniques are frequently limited by recurrence and distant metastasis. Recent pre-clinical and clinical studies have suggested that percutaneous ablative therapies lead to an alteration of the patient’s immuno-profile [16]. Among these immunological effects, for some therapies such as cryoablation and IRE, the central area of necrosis caused by the percutaneous ablation induces antigenic release that leads to an antigenic presentation by dendritic cells, increase in serum cytokines level, activation of the CTLA4 cascade and T cell response [17,18]. On the other hand, the peripheral area of apoptosis induced by ablation downregulates the immunological system [17]. These interactions produce both local and systemic effects, including occasionally the abscopal effect of distant tumor shrinkage [6,11,17,19,20].

While the immune response induced by ablation alone appears to be transient, there is strong evidence that it could potentially enhance the effect of immunotherapies [16,19,21,22,23,24,25,26,27,28,29,30,31,32,33,34,35,36,37,38,39]. In this section, we review the evidence regarding the most common percutaneous interventional techniques and their interaction with immunotherapies in cancer treatment.

### 2.1. Radiofrequency Ablation (RFA)

A sufficiently high thermal insult induces coagulation necrosis in the target tissue and cytokines and antigens blood release, leading to both local and systemic effects [40]. Slovak et al. demonstrated that, in a VX-2 rabbit liver cancer model, the combination of RFA plus CpG-B (a factor that stimulates innate immunity) increased the presence of activating lymphocyte and the rabbit survival compared with either RFA or CpG-B alone [41].

Schneider et al. analyzed the ablated area in non-small cell lung cancer (NSCLC), demonstrating a surge of CD4+ and CD8+ lymphocytes in the peripheral zone and an intensification of pro-inflammatory cytokines [42].

Mizukoshi et al. investigated the immune responses before and after RFA in 69 HCC patients and highlighted that there was a significant increase in tumor-associated antigen (TAA)-specific T cells in the peripheral blood of 62.3% of patients. Moreover, the number of TAA-specific T cells after RFA was predictive of HCC recurrence after ablation by univariate and multivariate analyses [43].

A comparative study involving patients with intermediate to advanced stage HCC investigated the efficacy of RFA plus monoclonal antibody (131I-chTNT) as a combination therapy [44]. This study demonstrated that such combination therapy is significantly more effective than RFA alone, as demonstrated by the longer survival time of patients who received RFA plus 131I-chTNT compared to those who received the RFA alone (*p* = 0.052) [44,45].

Regarding the systemic abscopal effects of radiofrequency ablation, it has been demonstrated in a colon-cancer murine model that the combination of RFA with a vaccine encoding CEA produces regression of distal metastasis and a significant increase in CEA-specific CD4+ T cells compared with RFA or vaccine alone (*p* < 0.0001; *p* = 0.0003, respectively) [46]. Nakagawa et al. demonstrated that the administration of dendritic cells stimulated by OK-432 (a clinical bacterial product that can induce DC maturation) after RFA increased the number of CD8+ T cells infiltrating untreated secondary tumors as compared to RFA alone (*p* < 0.001) [47]. However, the abscopal effect achieved by the RFA alone is weak, transient, or even occasionally counterproductive [41]. In fact, there is the risk of inducing an immunologically tolerogenic state if the RFA is not supported by immunotherapy. Indeed, it has been shown in rat breast cancer that RFA alone stimulates hepatocyte growth factor (HGF) and vascular endothelial growth factor (VEGF), leading to unwanted effects, such as an increased cell replication (evaluated by Ki-67) and microvascular density in distant tumors [48]. Therefore, the deactivation of HGF (using PHA-665752) and VEGF (using semaxanib) pathways may improve clinical outcomes of RFA ablation [48]. This opposite tumorigenic effect of RFA may also explain a worse prognosis of ablated HCC compared with surgical resection. Other clinical consequences of this pro-tumorigenic effect are the evidence that an incomplete radiofrequency ablation enhances neo-angiogenesis in HCC and tumoral growth in non-small cell lung cancer [49].

### 2.2. Cryoablation

Cryoablation is based on a cycle of freezing and thawing that causes intra- and extracellular ice crystal formation, damage to the cell membrane, osmotic pressure changes, and, thus, cellular dehydration. The use of cooling energy makes cryoablation suitable for lesions close to vital structures [50]. While the central area of the ablation is composed of necrotic tissue, the peripheral boundary is largely composed of apoptotic cells [51].

Despite all ablation techniques releasing tumor antigens, cryoablation avoids protein denaturation and preserves native antigen structures [52,53]. As a consequence, serum levels of interleukin-1 (IL-1), IL-6, NF-κβ, and TNF-α are significantly higher after cryoablation compared to other ablative therapies, suggesting a stronger immunostimulatory response [19].

In a renal cell carcinoma model, the combination of cryoablation and anti-PDL1 drug lead to anti-tumor immune responses and delayed tumor growth of distant untreated tumors [54]. Furthermore, in a melanoma model, den Brok et al. demonstrated that the combination of cryoablation and CpG-B induced the regression of the existing secondary tumors in 40% of cryoablation-treated mice, suggesting strong abscopal effects [19].

In metastatic liver cancer patients, Niu et al. demonstrated that the combination of cryoablation and immunotherapy leads to a significantly increased median overall survival (OS) compared with cryoablation or immunotherapy alone (32 vs. 17.5 vs. 3 months; *p* < 0.05) [25]. Similar results in terms of OS and immune responses were reported in patients with lung, renal cell, and hepatocellular cancers treated with cryoablation and allogeneic NK cell transfers and in patients with breast cancer treated with cryoablation plus anti-CTLA4 and anti-PD1 [55,56,57,58]. Despite the volume and the level of evidence being lower for cryoablation with respect to RFA, the combination of cryoablation with immunotherapy appears to offer promising results representing an optimistic basis for further investigations.

### 2.3. Irreversible Electroporation (IRE)

IRE is a novel non-thermal ablation technology based on the application of pulsatile and targeted high-voltage electric energy that alters the current potential of the cellular membrane, leading to permanent nanopore formation within the lipid bilayer membrane. This membranous disruption results in loss of homeostasis with subsequent cellular apoptosis and death.

The first significant evidence of the immunological effects of IRE was reported in 2016 by Bulvik et al., who demonstrated a greater lymphocyte infiltration and tumor size reduction for IRE compared to RFA in an HCC murine model [59]. Furthermore, in preclinical models of hepatocellular carcinoma, Vivas et al. showed that the administration of an immunostimulant drug (Poly-ICLC) before IRE was able to increase the immunogenic response and reduce tumor growth compared to both IRE and Poly-ICLC alone (40%, *p* < 0.05) [60]. These findings were confirmed by Alnagger et al., who reported an increased median overall survival (10.1 months of the IRE-NK group vs. 8.9 months of the IRE alone group, *p* = 0.0078) and a decrease in alpha-fetoprotein expression in patients with metastatic liver tumor (IV stage) treated with IRE plus allogeneic NK cell immunotherapy [61]. The same strategy (IRE plus NK vs. IRE alone) was investigated by Yang et al., who demonstrated longer median progression-free survival (PFS) and overall survival (OS) (PFS 15.1 vs. 10.6 months, *p* < 0.05, OS 17.9 vs. 23.2 months, *p* < 0.05) with a reduction of circulating tumor cells in patients who received combination therapy [62].

IRE has also been evaluated in other clinical contexts, as classical systemic immunotherapy has only limited efficacy against pancreatic ductal adenocarcinoma (PDAC) due to the presence of an immunosuppressive tumor-associated stroma, and the rationale of studies on IRE is that ablative therapies could destroy the pancreatic immunosuppressive microenvironment, leading to a greater response to systemic immunotherapy [63,64]. Zhao et al. utilized a mouse model of PDAC to demonstrate that the association of IRE and systemic anti-PD1 treatment promotes CD8+ T cell infiltration and increases overall survival when compared to both IRE and anti-PD1 as monotherapy [64]. Narayanan et al. also utilized a mouse model of PDAC. They combined IRE with systemic anti-PD1 and an intra-tumoral TLR7 agonist. This triple strategy improved local response compared to IRE alone and promoted regression of untreated concomitant metastases [65]. These encouraging results have brought to first preliminary human studies, showing that IRE combined with NK cells or allogenic Vγ9Vδ2 T cell infusion has prolonging effects on progression-free survival rates (11 versus 8.5 months), overall response rates at 1 month, and overall survival rates (14.5 versus 11 months) when compared to IRE alone in PDAC patients [66,67,68]. The soon-to-be-conducted PANFIRE-III trial (NCT04612530) will also combine IRE, systemic anti-PD1, and an intra-tumoral TLR9 agonist in metastasized PDAC human patients.

### 2.4. Microwave Ablation (MWA)

The association of MWA with immunotherapy is weaker, as preliminary studies suggested that MWA is less immunogenic compared to RFA and cryoablation [69]. However, Leutche et al. uncovered de-novo or enhanced tumor-specific T-cell responses in 30% of patients with hepatocellular carcinoma (HCC) treated with MWA alone. The T-cell response was associated with longer progression-free survival (27.5 vs. 10.0 months) [6]. In the same study, the analysis of HCC samples (n = 18) of patients receiving combined MWA and resection revealed superior disease-free survival in patients with high T-cell sample infiltration at the time of thermal ablation (37.4 vs. 13.1 months).

Regarding the synergic effect of MWA and immunotherapy, Chen et al. demonstrated that the combination of MW and GM-CSF significantly increased the free tumor survival and decreased the tumor volume in a murine hepatoma model [70]. Similar results were obtained in human patients with HCC, although in this initial study, the increase was not statistically significant [71]. Additionally, in a pilot study by Zhou P et al., the application of adoptive immunotherapy in association with MWA for HCC patients was demonstrated to be safe and capable of increasing the percentage of peripheral lymphocytes [72].

### 2.5. High-Intensity Focal Ultrasound (HIFU) and Laser-Induced Thermotherapy (LiTT)

Less evidence can be found in the literature for other ablation techniques. HIFU has been used for primary and secondary malignancy of the breast, soft tissue, bone, pancreas, kidney, and liver [6]. Yet, although HIFU can induce cytokine release and stress response with an augmented CD4+/CD8+ ratio, it appears to be less immunogenic compared with RFA and cryoablation [69].

LiTT has been reported to increase the level of cytokines (IL-6, TNFRI, and CRP levels) in liver malignancies [73]. Moreover, Vogl et al. highlighted that the levels of CD3+, CD4+, and CD8+ were increased after LiTT (12.73 ± 4.83 vs. 92.09 ± 12.04; 4.36 ± 3.32 vs. 42.92 ± 16.68; 3.64 ± 1.77 vs. 47.54 ± 15.68; *p* < 0.05) with an associated improvement in cytotoxic effects (RLU = 1493 ± 1954.68 vs. 7260 ± 3929.76; *p* < 0.001) [74].

Ablative techniques associated with immunotherapy seem to obtain a synergistic effect, as ablative therapies alone can increase neoangiogenesis when complete ablation is not achieved, also leading to immune tolerance. Most of the literature studies were made on RFA, whereas the combination of immunotherapy with cryoablation, MWA, IRE, and HIFU is emerging as a promising alternative to RFA on a great variety of target lesions. Table 1 summarizes the pros and cons of the current practice of locoregional percutaneous interventional oncology treatments when associated with immunotherapy, also reporting results of preclinical studies.

Table 2 describes in what lesions the clinical trials investigated the role of immunotherapy associated with ablative treatments.

## 3. Endovascular Treatments and Immuno-Oncology

In current practice, interventional intra-arterial treatments of tumors include the use of a wide variety of active tumoricidal agents, including conventional transcatheter arterial chemoembolization (cTACE), drug-eluting beads transcatheter arterial chemoembolization (DEB-TACE) and transarterial radioembolization (TARE) [75,76,77].

Chemoembolization (both cTACE and DEB-TACE) is the treatment of choice in patients with intermediate-stage HCC (Barcelona Clinic Liver Cancer—BCLC stage B), while radioembolization can potentially be used as an alternative based on level 2 of evidence in these patients [78,79,80]. Moreover, there is the ever-increasing use of these techniques for the treatment of hepatic metastases, including colorectal and neuroendocrine cancer [81,82]. In this section, we review the evidence of interaction between immunotherapy and the most common endovascular interventional techniques.

### 3.1. Transarterial Chemoembolization (TACE)

TACE is commonly used in patients with unresectable HCC with preserved liver function [3,78,83]. Conventional TACE (cTACE) commonly delivers an emulsion of lipiodol and chemotherapeutic agent (most often doxorubicin or cisplatin) followed by gelatine sponge as the embolic agent, whereas the most recent drug-eluting beads (DEB) TACE utilizes drug-eluting beads preloaded with the chemotherapeutic agent, with a reported reduction of drug-related side effects due to a better pharmacokinetic profile [83,84].

Both types of TACE induce local tumor necrosis by the occlusion of feeding arteries, leading secondarily to the release of tumor antigens, which activate the immune response [10]. Moreover, TACE can potentially modify the cytokine spectrum and the activation level of T cells [85]. TACE stimulates the secretion interleukines (IL) as IL-1 and IL-10, and of Interferon-γ, with activation of T helper-17 and T helper-1 cells [8,86,87]. TACE also leads to a modulation of immunosuppressive factors such as T-regulatory cells, PD1/PDL1, and HIF-1α, potentially bringing immune tolerance [87,88,89].

The combination of TACE and immunotherapy may amplify the antitumoral effect. In a pilot study, Sangro et al. used a CTLA4 inhibitor (tremelimumab) combined with TACE in 21 advanced HCC patients, showing promising results, with a good safety profile and a median survival of 8 months. Tremelimumab was administered intravenously at a dose of 15 mg/kg every 90 days until progression or intolerable toxicity [90,91].

Duffy et al. showed the efficacy of combined treatment of TACE/ablation and tremelimumab in a group of 32 patients with advanced HCC (75% of patients had the progressive disease). The median overall survival was 12.3 months, and most patients showed a reduction of tumor load, tumor reduction in non-ablated or non-embolized areas, and intratumoral infiltration of CD8+ T cells [92].

A phase-I clinical trial (NCT 03143270) in patients with advanced HCC treated with nivolumab (a PD1 inhibitor) combined with DEB-TACE is ongoing. All patients are scheduled to receive 24 mg of nivolumab intravenously for up to 1 year every 2 weeks.

Another open-label single-arm phase II study (NCT 03572582) combines nivolumab with TACE in patients with intermediate HCC is ongoing. Nivolumab treatment will start 2–3 days after the initial TACE and will be administered intravenously (240 mg, fixed-dose) every 2 weeks for up to two years until progression. A second TACE will be performed 8 weeks after the first one.

Other investigations using durvalumab (a PDL1 inhibitor) plus tremelimumab combined with TACE are underway, including a phase-II clinical trial (NCT02821754) and a clinical study (NCT03638141).

### 3.2. Transarterial Radioembolization (TARE)

TARE is emerging as a “multi-purpose” treatment in patients with HCC, as it can represent an effective alternative to both TACE and, as more recent studies demonstrate, ablative treatments. Under ideal conditions, it can lead to possible tumor downstaging and also act as a bridge to surgical resection and liver transplantation in selected patients [3,93,94].

TARE is usually performed with Yttrium-90 (90Y) in resin or glass microspheres (although there are ever-increasing reports regarding the effectiveness of Holmium-166 (166Ho) in poly(L-lactic acid) (PLLA) microspheres) [95]. As opposed to TACE, TARE utilizes local beta radiation to achieve tumor necrosis instead of artery occlusion. It is the following two-step treatment: where the first part is comprised of pre-treatment angiography with an injection of macroaggregated albumin (MAA) marked with Technetium-99m and a scintigraphy in order to evaluate lung shunt fraction (to avoid the risk of radiation pneumonitis) and to identify arteries that supply the gastrointestinal tract (to avoid ulcerations). The second part is the actual treatment with 90Y-loaded microspheres [96]. When judiciously delivered, TARE has a minimal embolic effect, so the post-embolization syndrome is reduced compared to TACE [97].

Recent studies underline that the immunocompetence of the tumor microenvironment is elevated after 90Y TARE. This effect is possibly explained by the expression of TNF-α by CD4+ T cells for the upregulation of CD8+ T cells, and the APCs ratio is increased [98].

A recent retrospective study of 26 patients with aggressive intermediate-stage or advanced HCC showed that the combination of nivolumab (or nivolumab plus ipilimumab) plus TARE is safe (little treatment toxicity) and has promising results, with a median overall survival of 16.5 months and progression-free survival of 5.7 months. One patient even achieved a complete response [99].

Moreover, a case report has been published of a patient affected by advanced HCC with a macrovascular invasion that was treated with nivolumab plus TARE, obtaining a downstaging and was amenable to surgery (with surgery confirming a complete response) [100].

These results are confirmed by a phase-II trial with 36 patients with an advanced HCC treated with TARE plus nivolumab, showing an overall response rate (31%), and an overall survival of 15.1 months [101].

Other trials are still evaluating TARE plus Nivolumab (NCT03380130, NCT03033446, NCT02837029) and TARE plus Pembrolizumab (NCT03099564).

The low volume and level of evidence of studies on combination treatments of immunotherapy and TACE or TARE make it difficult to discuss its safety and long-term efficacy, even though phase-I and -II clinical trials are ongoing and will certainly shed light on this promising combination technique. Table 3 summarizes the pros and cons of the current practice of locoregional endovascular interventional oncology treatments when associated with immunotherapy, also reporting results of preclinical studies.

Table 4 describes in what lesions the clinical trials investigated the role of immunotherapy associated with endovascular treatments.

## 4. Conclusions

The combination of interventional radiology treatments with immunotherapies in oncology is growing rapidly. The results, albeit currently with only a low level of evidence, are encouraging, suggesting that more evidence could better define the role of combination therapies in this field. In particular, the greatest evidence is focused on the combination of immunotherapy with RFA or cryoablation, whereas the combination of IRE and immunotherapy may play a greater role in the future, particularly in patients with pancreatic malignancies, where other methods have shown reduced efficacy.

The evidence from the literature is still rather limited for TACE and TARE, composed mostly of small monocentric studies. Further research in this field is required, ideally with randomized trials, to better understand how to achieve the desired immunogenic/abscopal effects while limiting unwanted pro-tumorigenic phenomena.

## Figures and Tables

**Figure 1 ijms-24-07344-f001:**
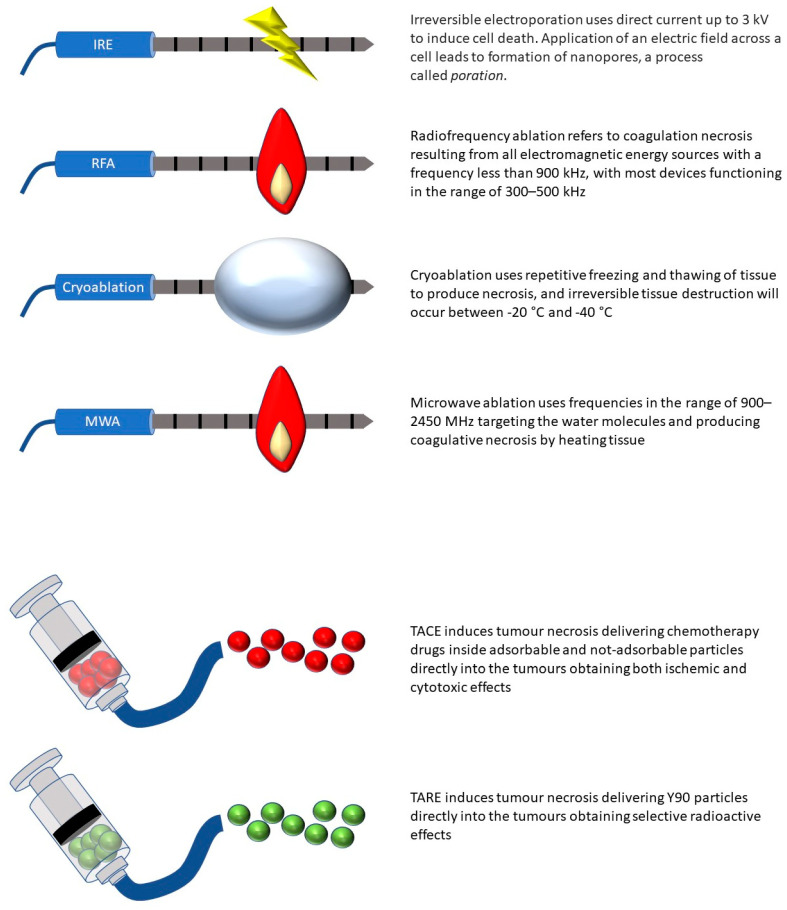
Interventional oncology ablative and endovascular techniques and mode of action. IRE: irreversible electroporation; RFA: radiofrequency ablation; MWA: microwave ablation; TACE: transcatheter arterial chemoembolization; TARE: transcatheter arterial radioembolization.

**Table 1 ijms-24-07344-t001:** Pros and cons of locoregional percutaneous interventional oncology treatments plus immunotherapy.

Technique	PROS	CONS
RFA	RFA + CpG-B increase activating lymphocytes and OS [41] RFA + monoclonal antibody increase OS [44,45] RFA + a vaccine encoding CEA produce regression of distal metastasis and increases CEA-specific CD4+ T cells [46] RFA + dendritic cells stimulated by OK-432 increase the number of CD8+ T cells infiltrating untreated secondary tumors [47]	RFA alone stimulates HGF and VEGF, increases microvascular density and tumor cell replication [48] Incomplete ablation enhances neo-angiogenesis and tumor growth [49]
Cryoablation	CA + anti-PDL1 drugs lead to anti-tumor immune responses and delayed tumor growth of distant untreated tumors [54] CA + CpG-B induces regression of existing secondary tumors [19] CA + immunotherapy increases OS [25] CA + allogeneic NK cells increase OS [55,56,57] CA + anti-CTLA4 and anti-PD1 increase OS [58]	Low volume and level of literature evidence compared to RFA
IRE	IRE + poly-ICLC increases immunogenic response and reduces tumor growth [60] IRE + allogeneic NK cell immunotherapy increases OS, PFS, and decreases AFP [61,62] IRE + anti-PD1 drug promotes CD8+ T cell infiltration and increases OS [64] IRE + anti-PD1 and TLR-7 agonist improves local response and regression of untreated lesions [65] IRE + NK cells or allogenic Vγ9Vδ2 T cell infusion improves PFS, OR, OS rates [66,67,68]	Low volume and level of literature evidence compared to RFA
MWA	MWA + GM-CSF increases DFS and decreases tumor volume [70] MWA + adoptive immunotherapy increases peripheral lymphocytes [72]	MWA seems to be less immunogenic compared to RFA and CA [69]
HIFU/LiTT	HIFU and LiTT increase cytokine levels [69,73,74]	HIFU induces less immunogenic effect compared with RFA and cryoablation [69]

RFA: radiofrequency ablation; OS: overall survival; CEA: carcinoembryonic antigen; CA: cryoablation; NK: natural killer; IRE: irreversible electroporation; PFS: progression-free survival; AFP: alpha-fetoprotein; OR: overall response; MWA: microwave ablation; DFS: disease-free survival; HIFU: high-intensity focused ultrasound; LiTT: laser-induced thermal therapy.

**Table 2 ijms-24-07344-t002:** Ablative technique, immunotherapy agents, and target lesions.

Technique	Immunotherapy Agent	Target Lesion and Study Type
RFA	Monoclonal antibody [44]	Intermediate- to advanced-stage HCC in liver cancer murine model study [44].
	CEA-encoding vaccine [46]	Distal colorectal cancer metastasis murine model study [46]
	OK-432-stimulated dendritic cells transfer [47]	In vivo untreated secondary tumors [47]
CA	Anti-PDL1 [54]	Distant untreated tumors in renal cell carcinoma murine model study [54]
	CpG-B [19]	Secondary tumors in melanoma murine model study [19]
	Immunotherapy	Metastatic liver cancer patients [25]
	Allogeneic NK cells [55,56,57]	Lung cancer, renal cancer, or HCC patients [55,56,57]
	Anti-CTLA4 and anti-PD1 [58]	Breast cancer patients [58]
IRE	Poly-ICLC [60]	Mice and rabbit HCC model study [60]
	Allogeneic NK cells [61,62]	Patients with metastatic liver tumor [61,62]
	Anti-PD1 [64]	Pancreatic ductal adenocarcinoma [64]
	Anti-PD1 and TLR-7 agonist [65]	Pancreatic ductal adenocarcinoma murine model [65]
	NK cells or allogeneic Vγ9Vδ2 T cell infusion [66,67,68]	Patients with pancreatic ductal adenocarcinoma [66,67,68]
MWA	GM-CSF [70]	Murine hepatoma model [70]
	Adoptive immunotherapy [72]	HCC patients [72]

RFA: radiofrequency ablation; HCC: hepatocellular carcinoma; CEA: carcinoembryonic antigen; CA: cryoablation; NK: natural killer; IRE: irreversible electroporation; MWA: microwave ablation.

**Table 3 ijms-24-07344-t003:** Pros and cons of locoregional endovascular interventional oncology treatments plus immunotherapy.

Technique	PROS	CONS
TACE	TACE + CTLA4 inhibitor increases OS [91,92] TACE + ablation and tremelimumab increases OS, reduces tumor load, reduces non-ablated or non-embolized tumors [93]	Low volume and level of literature evidence Phase-I and -II clinical trials ongoing
TARE	TARE + nivolumab (or nivolumab + ipilimumab) showed good results in terms of OS and PFS [100]	Low volume and level of literature evidence Phase-I and -II clinical trials ongoing

TACE: transarterial chemoembolization; OS: overall survival; TARE: transarterial radioembolization; PFS: progression-free survival.

**Table 4 ijms-24-07344-t004:** Endovascular technique, immunotherapy agents, and target lesions.

Technique	Immunotherapy Agent	Target Lesion and Study Type
TACE	CTLA4 inhibitor [91,92]	Advanced HCC patients [91,92]
TACE + ablation	Tremelimumab [93]	Advanced HCC patients [93]
TARE	Nivolumab [100]	Retrospective study on advanced HCC patients [100]

TACE: transarterial chemoembolization; HCC: hepatocellular carcinoma; TARE: transarterial radioembolization.

## Data Availability

No new data were created or analyzed in this study. Data sharing is not applicable to this article.

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
