# Peer review of "Interventional Oncology and Immuno-Oncology: Current Challenges and Future Trends"

_ijms, 2023, doi:10.3390/ijms24087344_

Round 1

Reviewer 1 Report

1.       Critical clinical trials related to interventional oncology and immuno-oncology should be summarized in more detail in tables, including, for example, information such as clinical trial timing, progress, results, etc.

2.       The current shortcomings of interventional oncology, particularly those that provoke an immune response, should be introduced.

3.       The results of the studies listed should be properly discussed and not simply listed.

Author Response

Q: Critical clinical trials related to interventional oncology and immuno-oncology should be summarized in more detail in tables, including, for example, information such as clinical trial timing, progress, results, etc.

A: Thank you for your comment: we added two more tables highlighting the target lesions and study cohort of combination therapies with immunotherapy and interventional radiology.

Q: The current shortcomings of interventional oncology, particularly those that provoke an immune response, should be introduced.

A: Thank you for the interesting comment: in both section on ablative and endovascular treatments, we describe the drawbacks of interventional radiology therapies, as they can lead to immune tolerance, immunosuppression, and tumor growth, thus underlining the importance and usefulness of their association with immunotherapy.

Q: The results of the studies listed should be properly discussed and not simply listed.

A: Thank you for your comment. As literature evidences are scarce, and mostly based on monocentric studies, we know that drawing conclusive statements can be a hard task; however, we added a few lines of text at the end of each section on both ablative and endovascular treatments to summarize the state of the art on combination of immunotherapy and interventional oncology procedures, also hinting to the final paragraph of the manuscript in which conclusions are drawn.

Reviewer 2 Report

ijms-2337570

Interventional Oncology and Immuno-Oncology: current challenges and future trends. 

The article “Interventional Oncology and Immuno-Oncology: current challenges and future trends. (ijms-2337570)” by Posa A, et al. demonstrated that interventional radiology treatment combined with immune therapy was effective and their therapeutic efficacy was dependent of tumor microenvironment according to previous reports, suggesting that these treatment strategy could be promising in the future. This review article was very interesting and encouraging for a lot of readers. However, the impact of this manuscript was relatively low because impressive figure was not present. Therefore, I suggested several points for the purpose to improving your review article.

1. I recommended that the figure concerning the mode of action for interventional treatment should be added.

2. What kinds of malignancies were the interventional treatment active for? The author could add that point using tables about the results of clinical trials.   

Author Response

Q: the impact of this manuscript was relatively low because impressive figure was not present. Therefore, I suggested several points for the purpose to improving your review article. I recommended that the figure concerning the mode of action for interventional treatment should be added.

A: Thank you for your suggestion. We added a figure highlighting the mode of action of all the interventional radiology treatments we talk of in the manuscript.

Q: What kinds of malignancies were the interventional treatment active for? The author could add that point using tables about the results of clinical trials.

A: Thank you for the suggestion. We added two more tables highlighting the target lesions and study cohort of combination therapies with immunotherapy and interventional radiology.